# The Effect of a Multifaceted Intervention on Dietary Quality in Schoolchildren and the Mediating Effect of Dietary Quality between Intervention and Changes in Adiposity Indicators: A Cluster Randomized Controlled Trial

**DOI:** 10.3390/nu14163272

**Published:** 2022-08-10

**Authors:** Jin-Lang Lyu, Zheng Liu, Shuang Zhou, Xiang-Xian Feng, Yi Lin, Ai-Yu Gao, Fang Zhang, Li Li, Antje Hebestreit, Hai-Jun Wang

**Affiliations:** 1Department of Maternal and Child Health, School of Public Health, Peking University, Beijing 100191, China; 2National Health Commission Key Laboratory of Reproductive Health, Peking University, Beijing 100191, China; 3Department of Preventive Medicine, Changzhi Medical College, Changzhi 046000, China; 4Urumuqi Primary and Secondary School Health Care Center, Urumuqi 830000, China; 5Dongcheng Primary and Secondary School Health Care Center, Beijing 100010, China; 6Mentougou Primary and Secondary School Health Care Center, Beijing 102300, China; 7Department of Endocrinology and Metabolism, Ningbo First Hospital, Ningbo 315000, China; 8Leibniz Institute for Prevention Research and Epidemiology-BIPS, 28359 Bremen, Germany

**Keywords:** children, obesity, intervention, dietary quality, sugar-sweetened beverage

## Abstract

Some studies have found associations between dietary quality and obesity and their concurrent changes were observed in a few interventions. The present study aimed to assess the effect of a multifaceted intervention for childhood obesity on dietary quality and examine the mediating effect of dietary quality between the intervention and changes in adiposity indicators. Based on the social ecological model, the cluster randomized controlled trial included five components (three targeted children and two targeted their environment). In total, 1176 children from three cities in China participated in a baseline (2018) and end-of-trial (2019) examination, including 605 children in the intervention group and 571 in the control group. Self-reported behavior and anthropometric measures were collected at both time points. The Diet Balance Index Revision (DBI-07) was calculated to assess dietary quality. Generalized linear mixed models were used to estimate the intervention effect on dietary quality and its mediating effects were examined. Compared to the controls, the proportion of sugar-sweetened beverage (SSB) intake (OR = 0.27, *p* < 0.001, corrected *p* < 0.001) decreased in the intervention group. Higher bound scores (HBS) of the DBI-07 indicating over-intake decreased in the intervention group compared to the controls (mean difference = −1.52, *p* = 0.005, corrected *p* = 0.015). Changes in the HBS partially mediated the associations between the intervention and changes in body mass index, waist circumference, and body fat percentage. Future intervention should promote knowledge, attitudes, and behaviors related to dietary quality.

## 1. Introduction

In the last four decades, the number of children and adolescents with obesity has increased 10-fold over the world, with the global age-standardized prevalence of obesity in the age group of 5–19 years increasing 8-fold [1]. Rapid increases have also been reported in China, as the prevalence of obesity among children and adolescents aged 6–17 years rose from 1.8% in 1992 to 7.9% in 2019 [2]. As childhood obesity could cause adverse health outcomes such as diabetes, hypertension, non-alcoholic fatty liver disease, and psychosocial complications [3], it has become a serious public health problem in China, and intervention measures of prevention and control are urgently needed.

Our research group conducted a systematic review and found that multicomponent interventions appeared to be more effective than single-component interventions and the effects of physical activity interventions including curricular sessions were stronger [4]. There is a scarce number of studies including dietary improvement components, and limiting subgroup analyses and comparisons between China and Western countries. Despite the assumption that targeting dietary improvement would promote a beneficial effect on childhood obesity, previous evidence also has inconsistent conclusions on its contribution [5,6,7,8].

Overall, dietary quality was more closely associated with childhood obesity because foods were not eaten in isolation and might have a cumulative impact [9]. The Dietary Quality Index, which could reveal the holistic dietary situation and diversity rather than merely the intake of individual foods or nutrients, was one of the most common methods to evaluate overall dietary quality in children [10,11]. Studies in China and abroad have provided an important foundation, as significant associations between dietary quality indices and obesity were found [12,13,14,15], and concurrent changes in them were observed in interventions in other countries [16,17,18]. However, direct evidence is scarce for the effect of dietary quality as an underlying mechanism (that is, a mediator) of a successful intervention with favorable changes in adiposity indicators in children. Considering China’s different dietary characteristics and patterns, it is necessary to study the mediating effect of dietary quality and add evidence from China.

Our research group recently published a cluster randomized clinical trial of an obesity intervention, which showed that a multifaceted intervention (DECIDE-Children) effectively reduced the BMI and obesity prevalence in primary school children [19]. Although improving the diet was an important component in the intervention, it was unclear whether it acted as a significant mediator of the intervention effect. Understanding the mechanisms behind the beneficial effect of an intervention can help to prompt future intervention developers to strengthen the effective components and redesign the ineffective ones [20]. Mediation analysis was used as an important statistical tool in previous successful obesity interventions to test the hypothesized underlying mechanisms, but previous studies had inconsistent results on the effect of dietary factors [6,7,8]. Thus, embedded within our primary research, which mainly explored the intervention’s effect on obesity prevention, the present study limited the focus to the improvement of dietary quality and its hypothesized mediating role. The objectives of our study included: (1) to assess the effects of a multifaceted intervention on dietary quality; and (2) to examine the mediating effect of dietary quality between the intervention and changes in adiposity indicators.

## 2. Methods

### 2.1. Study Design and Participants

The Diet, ExerCIse and CarDiovascular hEalth Children (DECIDE-Children) study was a cluster randomized controlled trial conducted across three socioeconomically distinct Chinese areas: Beijing, Changzhi of Shanxi Province, and Urumuqi of Xinjiang Province. A total of 24 schools (8 schools in each region) were randomly assigned at a 1:1 ratio to either the intervention or the control group after baseline measurements. Randomization was stratified by district (urban or suburban) within each area and performed by an independent person at the Clinical Research Institute of Peking University using a computer-generated random number system. The children of Grade 4 from each primary school were recruited. A full trial protocol has been published [21] and ethical approval was granted by the Peking University Institutional Review Board (IRB00001052-18021). In the current study, the children from the original study were excluded if they did not complete the dietary assessment at baseline or at the end of the trial, or if their reported dietary intake had missing values.

### 2.2. Intervention

The multifaceted intervention was developed based on the social ecological model. The duration of the intervention was one school year from September 2018 to June 2019. In brief, the intervention included five components: three targeting the children directly (health education activities, the reinforcement of physical activity during class break and physical education (PE) lessons, and monitoring of the children’s weight and height) and two targeting the children’s environments at school (e.g., instructing school principals/teachers and implementing school policies) and within the family (e.g., providing health education to parents and helping them to record their children’s behaviors and track feedback), respectively. A smartphone app was also used to facilitate communication between the intervention personnel and family participants. A more detailed description of the implemented intervention was provided and the results showed that schools and parents had relatively good compliance (Appendix A).

Interventions related to diet are listed as follows. Health education books including instructions on healthy diet and “Nutrition evaluation turnplate for Chinese primary and middle school students” were delivered to children for learning at school and home. Trained teachers taught children about methods of achieving a healthy weight (not eating excessively, not drinking sugar-sweetened beverages, and eating less high-energy food). School policies (e.g., putting up health education posters and not selling SSB or unhealthy snacks) were implemented. Parents together with their children were asked to record dietary behaviors weekly in the app and could obtain individualized feedback on how they performed according to the intervention targets.

### 2.3. Outcome Measures

#### 2.3.1. Assessment of Dietary Intake

Food consumption during the previous seven days was assessed at baseline and at the end of the trial using a Food Frequency Questionnaire (FFQ), which was a slightly revised version of a validated FFQ for children and adolescents developed in the study by Yan et al. [22,23]. After removing the questions about sauce, oil, and dietary supplements, the revised FFQ contained 12 food groups and their daily consumptions were calculated by multiplying the intake frequency per day by the amount consumed on each occasion. 

#### 2.3.2. Calculation of Dietary Quality

Dietary quality was assessed using the Diet Balance Index Revision (DBI-07), which was widely used in school-aged children in China [24,25,26,27]. Considering the differences in dietary characteristics between adults and children, and referring to the method of Duan et al. [25] and Zheng et al. [28], the DBI-07 was revised to meet the dietary requirements of the children in this study.

The revised DBI-07 consisted of 12 subgroups: (1) cereals; (2) vegetables; (3) fruits; (4) milk and dairy products; (5) soybean and soybean products; (6) red meat, poultry, and game; (7) fish and shrimp; (8) eggs; (9) sugar-sweetened beverages; (10) unhealthy snacks; (11) dietary diversity; and (12) drinking water (Appendix A). A score of 0 for each subgroup indicated that the individual’s food intake met the recommended amount by the Chinese Dietary Guidelines. Similar to the methods used in adults [29], positive scores were used to evaluate the over-intake of foods (SSB and unhealthy snacks) that were recommended to be consumed in a “reduced” or “limited” amount, according to the dietary guideline. Negative scores were used to evaluate the under-intake of foods (vegetables, fruits, milk and dairy products, soybean and soybean products, fish and shrimp, and drinking water) that were recommended to be consumed in a “sufficient” or “plenty” amount. Both positive and negative scores were used to evaluate the intake of foods (cereals, red meat, poultry and game, and eggs) that were recommended to be consumed in an “appropriate” amount. Dietary diversity was assessed by the total scores of the intakes of the 12 food subgroups collected by the FFQ. If the intake of a subgroup exceeded 25 g, a score of 1 was given, otherwise a score of 0 was assigned. Based on the scores evaluated, the intake of each food subgroup was divided into under-intake or over-intake as categorical variables, and dietary diversity was divided into adequate or inadequate.

Three indicators of overall dietary quality were calculated. Higher bound scores (HBS) were the sum of all the positive scores, indicating the overall over-intake. Lower bound scores (LBS) were the sum of the absolute values of all the negative scores, indicating the overall under-take. Diet quality distance (DQD) was calculated by adding the absolute values of all the positive and negative scores, indicating the overall unbalanced intake.

#### 2.3.3. Adiposity Indicators

The adiposity indicators in this study included body mass index (BMI), BMI z-score, waist circumference (WC), and body fat percentage (BF%). Children’s anthropometric measures at both time points were collected by trained personnel, using identical devices and standardized forms referring to standard methods and procedures. The BMI z-score was calculated according to World Health Organization (WHO) standards [30]. Obesity was evaluated using age- and sex-specific BMI percentiles according to the Chinese reference [31].

### 2.4. Covariates

Children’s demographic characteristics (sex and age) were obtained by questionnaires at baseline survey. Considering the associations between physical activity, screen time, and childhood obesity [32], changes in these behavior indicators were adjusted in the mediation analyses. Children’s duration of moderate-to-vigorous physical activity was collected by the validated International Physical Activity Questionnaire-Short Form (IPAQ-SF) [33]. Screen time was collected by a self-designed question (How much time on average did you spend watching television and playing on electronic devices every day during the last week?).

### 2.5. Statistical Analyses

Categorical variables were presented as numbers and percentages (%). Continuous variables with abnormal distribution were presented as medians with interquartile ranges (IQR). Basic characteristics were compared using the Wilcoxon Rank Sum Test for continuous data, with Chi-Square tests for categorical variables. Generalized linear mixed models were used to analyze the effects of the intervention on dichotomous outcomes (12 subgroups) and continuous outcomes (three dietary quality indicators). Children’s age, sex, region, and the corresponding dietary intake at baseline were adjusted, along with a school-level random intercept considering the correlation due to clustering of children within schools. The Benjamini–Hochberg procedure was used to control the false discovery rate for multiple comparisons for each outcome. For sensitivity analyses, we further adjusted for district (urbansuburban) in the models and imputed the missing values of those who were lost to follow-up at the end of the trial using baseline values. In the subgroup analyses, interaction terms between each subgroup variable and group assignment were used in the models to examine whether the intervention effects differed significantly by subgroup.

In the mediation analyses, three regression analyses were conducted [34]: The regression coefficients for the intervention effect on the change in dietary quality was determined in Path A. The association between dietary quality and adiposity indicators adjusted for the group assignment was determined in Path B. The regression coefficients for the intervention effect on the adiposity indicators directly (total effect) was determined in Path C, and the coefficients of the intervention on the adiposity indicators when adjusting for the change in dietary quality (direct effect) was determined in Path C’. All pathway models were adjusted for baseline dietary intake, adiposity indicators, and covariates, along with a school-level random intercept. When the mediator was a continuous variable with a normal distribution, products of coefficients in pathway A and B (a × b) represented the indirect effect, of which the estimate and 95% confidence intervals (CI) were calculated using the mediation package in R software [35]. The proportion of mediation was calculated as the ratio of the indirect effect to the total effect. If the mediator did not follow a normal distribution, traditional analysis methods based on normal theory would yield inaccurate confidence limits and significance tests. Thus, the significance of the indirect effect was tested using the PRODCLIN method [36]. The results were considered statistically significant at two-sided *p* < 0.05. Statistical analyses were carried out using R software (version 4.0.3; creator: John Chambers and colleagues; location: Jersey City, NJ, USA).

## 3. Results

### 3.1. Characteristics of Participants

A total of 1176 children who met the inclusion criteria were involved in this study (Figure 1). The children included did not differ from those who were excluded on demographic variables (Appendix A). The control group (*n* = 571) and intervention group (*n* = 605) had similar sociodemographic characteristics at baseline. There were no significant differences in other characteristics between the two groups (Table 1).

### 3.2. Effects of the Intervention on Indicators of Food Subgroups and Dietary Quality

Table 2 shows the levels of the 12 subgroups in the control and intervention groups at baseline and at the end of the trial. From baseline to the end of the trial, the prevalence of over-intake of SSB changed by −18.4% (intervention group) and 12.0% (control group), with an odds ratio of 0.27 (95% CI: 0.19, 0.40, *p* < 0.001, corrected *p* < 0.001). The changes in over-intake of unhealthy snacks in the intervention and control groups were −10.1% and 1.5%, respectively, with an odds ratio of 0.59 (95% CI: 0.37, 0.93, *p* = 0.023, corrected *p* = 0.172) before correcting for multiple comparisons. Differences in the changes in the levels of the food subgroups and dietary diversity were not statistically significant (*p* > 0.05). The result remained consistent when we further adjusted for district or imputed missing values at the end of the trial (see Appendix A).

Table 3 shows the differences in the changes in the three dietary quality indicators between the two groups. Notably, the HBS indicating over-intake decreased from 6.03 to 4.51 in the intervention group, while it increased from 5.28 to 5.89 in the controls. The mean difference in the changes between the two groups was −1.52 (95% CI: −2.42, −0.62, *p* = 0.005, corrected *p* = 0.015). Differences in the changes of the other indicators were not statistically significant (*p* > 0.05). The result remained consistent when we further adjusted for district or imputed missing values at the end of the trial (see Appendix A).

When stratified by sex, obesity status at baseline, and region, we did not observe significant differences of the improvement on the SSB or HBS scores. The effect size did not differ significantly by sex, baseline obesity status, and region. (Figure 2).

### 3.3. Mediation Analyses

The indirect effects of each of the 12 subgroups, LBS, and DQD were not statistically significant in this study. Increases in the HBS were associated with lower adiposity indicators. Changes in the HBS mediated 4.01% (DE = −0.37, *p* < 0.001; IE = −0.02, *p* = 0.026), 4.51% (DE = −1.40, *p* = 0.010; IE = −0.07, *p* = 0.030), and 5.67% (DE = −0.85, *p* = 0.008; IE = −0.05, *p* = 0.026) of the association between the intervention and changes in BMI, WC, and BF%, respectively (Table 4).

## 4. Discussion

The DECIDE-Children study demonstrated the effectiveness of a multifaceted intervention in obesity prevention. In the current study, this multifaceted intervention was also found to improve the dietary quality of children, and their SSB intake and over-intake scores (HBS) decreased after the intervention. To our knowledge, this is the first study that has examined overall dietary quality as a mediator in obesity intervention effects among children. Mediation analyses revealed that the decrease in over-intake had a mediating effect between the intervention and the changes in BMI, WC, and BF%, which filled the research gap.

The DECIDE-Children study effectively improved dietary performance in children. The intervention effect size for the HBS was −1.52, with a standard deviation (SD) of 0.48. When assuming an intra-cluster correlation coefficient of 0.05, a total of 1176 children from 24 schools (clusters) was estimated to provide a power of >90% at α = 0.05 [19]. Intervention procedures adapted to the cultural and dietary practices in China played an important role in determining intervention effectiveness. Firstly, as the previous study pointed out, prevention interventions appeared to be more effective in countries such as China where the obesity epidemic was at an earlier stage and fewer existing initiatives have been implemented by government, local authorities, or schools [37]. In addition, in China, most primary schools are government-funded public schools, and policies such as not selling sugary drinks and snacks in schools do not considerably affect school revenues and were easy to generalize. However, in Western countries, after eliminating vending machines and stores, many private schools would struggle with maintaining profit margins and finding suitable fundraising alternatives [38]. It is important to point out that we encouraged the engagement and cooperation of both families and schools by using apps or implementing school policies to create a nutritional environment. In China, parents’ meetings are held regularly in schools and Wechat groups were set up for parents and schoolteachers, which helped to provide opportunities to deliver health education. Chinese parents also gave more attention to advice from schoolteachers than from other social institutions, which may be different to other countries [39]. Zheng et al. [28] provided a regular healthy eating curriculum at school to primary school students along with their parents. Similar to our results, the Dietary Quality Index Higher Bound (DQI-HB) scores indicating over-intake in the interventions were significantly lower compared to the control group (15.4 vs. 21.9, *p* < 0.05). Moreover, participatory approaches combining family and school have also been proved effective in changing eating behavior [40,41]. Health education activities and policies at school, students learning nutritional concepts in the books and app with parents at home, and parents’ supervision of eating behaviors could increase the familiarity and use of nutrition knowledge among students [40]. In this study, the intake of other food subgroups was not significantly improved. Learning from other successful cases, future dietary interventions could consider peer education [42], giving more specific instructions [43], and improving the canteen food supply [37] to effectively and comprehensively enhance children’s dietary quality.

Dietary indicators of single foods (fruits, vegetables, SSB) in intervention effects on weight loss have been proved. An intervention in Spain analyzed the correlation of concurrent changes in the DQI and body indicators but did not give robust evidence of its contributions to the intervention effects. Their results showed that 5-unit increases in the DQI-A score resulted in a BMI z-score decrease of 0.07 and in an FMI z-score decrease of 0.053 [18]. Despite direct comparisons being impossible due to the use of different methods and diet quality indices, we found that higher quality was associated with better anthropometric outcomes. Although the effect size was not very large, behavior change in childhood might be paramount and accumulate over time to exert larger effects [7]. These findings also, to some extent, address the gap in the association of dietary quality and children’s adiposity outcomes as most of the previous studies had a cross-sectional design. Our intervention had no significant improving effects on other food groups. It is necessary to further explore feasible methods to improve other aspects of dietary intake. However, the children did make great progress in overall dietary quality, which means that just small changes in choosing more healthy foods instead of calorically dense nutrient-poor foods might add up to have a cumulative effect. Over-intake of refined cereals, animal foods, unhealthy snacks, and SSB would lead to an increase in the HBS of the DBI-07. These foods have been considered as potentially obesogenic foods and frequent ingestion was associated with the risk of obesity in observational studies [44]. Potential explanations included increasing the total energy intake and causing energy surplus [45,46]. Therefore, a decrease in the HBS exerted a positive effect on obesity by keeping energy and nutrient intake in balance. We provided important evidence that an improvement of dietary quality was a part of the mechanism underlying successful multifaceted intervention, and had great implications for future prevention and control of childhood obesity. In addition to emphasizing rational intake of fruits, vegetables, and SSB in the previous intervention, both the quality and quantity of the different foods should also be given sufficient attention.

A few challenges need to be overcome to minimize the effect’s reduction when scaling up the intervention in the future. Firstly, obesity prevention has not previously attracted enough attention from schools, as greater attention has been placed on academic performance by schools and families. The advancement of the State Council’s Action Plan for Healthy China, which included nutrition and health education for children, will help to promote widespread health education in schools [40]. Another challenge remaining is how to fully engage family support in school-based interventions. Parental involvement in childhood obesity interventions is important as children are highly influenced by their family unit. However, some school-based interventions showed difficulties in encouraging parental support. In the future, schools could consider various approaches, such as holding interactive classes and parent meetings, assigning homework, and monitoring BMI via mobile health technology, to increase the attention of parents to obesity prevention and blend the cooperation of family and school.

The main strengths of this study included exploring the effect of intervention on dietary quality and mediating effect analyses, which helped to understand how the multifaceted intervention affected childhood obesity, and for the first time, provided further evidence that changes in dietary quality are an important factor behind the beneficial effect of intervention. In the mediating analyses, we selected the BMI, BMI z-score, WC, and BF% to thoroughly reflect the children’s body size and composition. In addition, potential confounding factors as physical activity time and screen time were adjusted to reduce bias.

This study has some limitations to be acknowledged. First, the use of the FFQ in investigating children’s daily intake might result in inaccurate estimates and recall bias. However, children finished the questionnaire with the help of trained researchers and were provided with the unified standard tableware diagram and illustrative food pictures. Thus, the bias was reduced to a certain extent. Second, physical exercise time and screen time were self-reported, and lacked objective instrument measurement. However, as clarified in the primary study, changes in behavioral measures paralleled the changes in objective measures of adiposity [19]. Third, dietary information was collected immediately after the intervention, and we did not know whether the intervention effects on dietary quality could maintain and exert further changes in body indicators in the future. 

## 5. Conclusions

In conclusion, the multifaceted study effectively improved children’s dietary quality, which in turn led to beneficial changes in adiposity indicators. Our findings identified the partial mediating effect of dietary quality in childhood obesity intervention. Future intervention strategies for childhood obesity prevention should promote knowledge, attitudes, and behaviors related to dietary quality.

## Figures and Tables

**Figure 1 nutrients-14-03272-f001:**
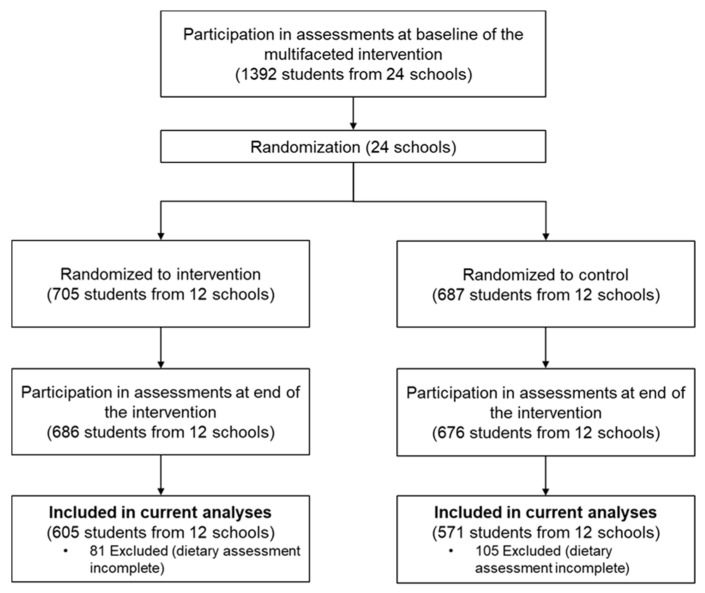
Flow Diagram of Selecting Study Participants.

**Figure 2 nutrients-14-03272-f002:**
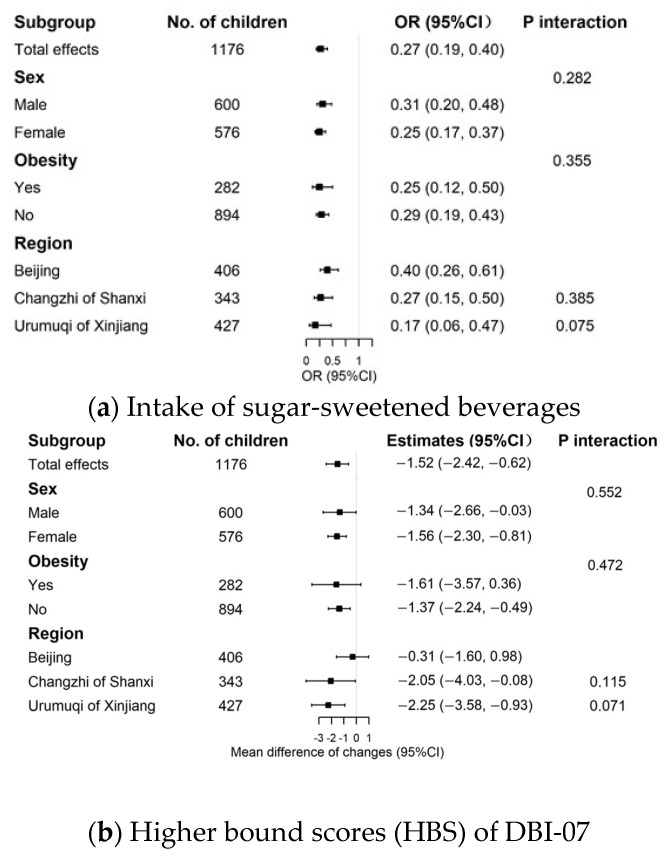
Subgroup analyses of the intervention effects on dietary quality. Figure (**a**) shows the subgroup analyses of the effects on sugar-sweetened beverage intake; figure (**b**) shows the subgroup analyses of the effects on HBS.

**Table 1 nutrients-14-03272-t001:** The characteristics of the two groups at baseline ^a^.

	Control (*n* = 571)	Intervention (*n* = 605)	*p*
**Cluster level**			
Number of schools	12	12	
Median (IQR) number of children/school	37 (10.0)	35 (9.5)	0.213
**Individual level**			
Sex, *n* (%)			0.585
Male	296 (51.8)	304 (50.2)	
Female	275 (48.2)	301 (49.8)	
Region, *n* (%)			0.749
Beijing	191 (33.4)	215 (35.5)	
Changzhi of Shanxi	170 (29.8)	173 (28.6)	
Urumuqi of Xinjiang	210 (36.8)	217 (35.9)	
Age (years)	9.61 (0.51)	9.62 (0.54)	0.875
Anthropometric measures			
BMI (kg/m^2^)	18.19 (5.43)	17.86 (4.97)	0.286
BMI z-score	0.82 (2.24)	0.74 (2.12)	0.265
WC (cm)	63.95 (16.02)	62.75 (14.00)	0.336
Body fat percentage (%)	19.40 (15.85)	18.70 (13.70)	0.482
Obesity ^b^, *n* (%)			0.406
No	428 (75.0)	466 (77.0)	
Yes	143 (25.0)	139 (23.0)	

^a^ Categorical variables were reported as *n* (%) and continuous variables were reported as medians (IQR). ^b^ Obesity was defined using age- and sex-specific BMI percentiles according to Chinese reference [31]. Abbreviations: BMI, body mass index; WC, waist circumference.

**Table 2 nutrients-14-03272-t002:** Effects of intervention on the levels of each subgroup ^a^.

	Control, *n* (%)		Intervention, *n* (%)		Intervention vs. Control	
	Baseline	End of Trial	Baseline	End of Trial	OR (95% CI)	*p*
**Under-intake**						
Cereals ^b^	398 (69.7)	396 (69.4)	418 (69.1)	451 (74.5)	1.33 (0.94, 1.88)	0.107
Vegetables	478 (83.7)	437 (76.5)	496 (82.0)	437 (72.2)	0.82 (0.54, 1.24)	0.346
Fruits	360 (63.0)	360 (63.0)	376 (62.1)	361 (59.7)	0.88 (0.65, 1.18)	0.394
Milk and dairy products	471(82.5)	422 (73.9)	490 (81.0)	461 (76.2)	1.15 (0.81, 1.63)	0.448
Soybean and soybean products	486 (85.1)	482 (84.4)	514 (85.0)	516 (85.3)	1.05 (0.75, 1.45)	0.781
Red meat, poultry, and game ^b^	325 (56.9)	268 (46.9)	335 (55.4)	307 (50.7)	1.19 (0.90, 1.57)	0.212
Fish and shrimp	516 (90.4)	521 (91.2)	547 (90.4)	556 (91.9)	1.07 (0.67 1.72)	0.780
Eggs ^b^	246 (43.1)	234 (41.0)	246 (40.7)	257 (42.5)	1.08 (0.69, 1.67)	0.755
Drinking water	433 (75.8)	385 (67.4)	430 (71.1)	368 (60.8)	0.73 (0.51, 1.03)	0.073
**Over-intake**						
Cereals ^b^	136 (23.8)	120 (21.0)	150 (24.8)	108 (17.9)	0.80 (0.55, 1.16)	0.246
Red meat, poultry, and game ^b^	141 (24.7)	167 (29.2)	152 (25.1)	172 (28.4)	0.95 (0.69, 1.31)	0.755
Eggs ^b^	220 (38.5)	229 (40.1)	239 (39.5)	245 (40.5)	1.08 (0.69, 1.67)	0.755
Sugar-sweetened beverages	207 (36.3)	276 (48.3)	249 (41.2)	138 (22.8)	0.27 (0.19, 0.40)	<0.001 *
Unhealthy snacks	82 (14.4)	91 (15.9)	125 (20.7)	64 (10.6)	0.59 (0.37, 0.93)	0.023 *
**Dietary diversity**						
Inadequate dietary diversity	559 (97.9)	560 (98.1)	592 (97.9)	593 (98.0)	0.97 (0.39, 2.42)	0.943

^a^ Generalized linear mixed models with logistic link function were used, allowing for the school clustering effect, with adjustment for age, sex, region, and the dietary outcomes at baseline. ^b^ Over-intake (score > 0) and under-intake (score < 0) situations were described for foods given both positive and negative scores in DBI-07. * *p* < 0.05.

**Table 3 nutrients-14-03272-t003:** Effects of intervention on three dietary quality indicators ^a^.

	Control, Mean (SD)		Intervention, Mean (SD)		Changes from Baseline (Intervention vs. Control)	
	Baseline	End of Trial	Baseline	End of Trial	Adjusted Mean Difference (95% CI)	*p*
Higher bound scores (HBS)	5.28 (5.70)	5.89 (5.86)	6.03 (6.04)	4.51 (5.37)	−1.52 (−2.42, −0.62)	0.005 *
Lower bound scores (LBS)	34.61 (13.45)	30.71 (12.25)	32.66 (12.62)	30.38 (12.45)	0.01 (−2.27, 2.28)	0.996
Diet quality distance (DQD)	39.90 (11.88)	36.60 (10.49)	38.68 (10.74)	34.88 (11.43)	−1.50 (−3.72, 0.70)	0.219

^a^ Linear mixed models were used, allowing for the school clustering effect, with adjustment for age, sex, region, and the corresponding scores at baseline. * *p* < 0.05.

**Table 4 nutrients-14-03272-t004:** Mediating role of dietary quality in the intervention effects on changes in BMI, BMI z-score, WC, and body fat percentage ^a^.

	BMI Change		BMI Z-Score Change		WC Change		BF% Change	
	Estimates (95% CI)	*p*	Estimates (95% CI)	*p*	Estimates (95% CI)	*p*	Estimates (95% CI)	*p*
**HBS ^a,b^**								
Direct effect	−0.37 (−0.59, −0.16)	<0.001 *	−0.14 (−0.22, −0.06)	0.002 *	−1.40 (−2.48, −0.33)	0.010 *	−0.85 (−1.45, −0.26)	0.008 *
Indirect effect	−0.02 (−0.04, 0.00)	0.026 *	−0.01 (−0.01, 0.00)	0.070	−0.07 (−0.16, 0.00)	0.030 *	−0.05 (−0.12, 0.00)	0.026 *
Total effect	−0.39 (−0.61, −0.18)	<0.001 *	−0.15 (−0.24, −0.06)	<0.001 *	−1.47 (−2.54, −0.42)	0.010 *	−0.90 (−1.50, −0.32)	0.006 *
Proportion of mediation, %	4.01 (0.31, 12.29)	0.028 *	3.46 (−0.15, 14.36)	0.070	4.51 (0.21, 17.10)	0.036 *	5.67 (0.43, 19.41)	0.028 *

^a^ Change in HBS as a mediator was a continuous variable. ^b^ A total of 62 (5.3%) missing values without screen time and moderate-to-vigorous physical activity time. * *p* < 0.05. Abbreviations: BMI, body mass index; WC, waist circumference; BF%, body fat percentage.

## Data Availability

Data described in the manuscript, codebook, and analytic code will not be made available because the Peking University Institutional Review Board has not consented to this. In order to access more information on data analyses, contact the corresponding author at whjun@pku.edu.cn.

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
