# Peer review of "The Effect of a Multifaceted Intervention on Dietary Quality in Schoolchildren and the Mediating Effect of Dietary Quality between Intervention and Changes in Adiposity Indicators: A Cluster Randomized Controlled Trial"

_nutrients, 2022, doi:10.3390/nu14163272_

Round 1
Reviewer 1 Report
The Authors performed a nice trial to assess the effect of a multifaceted intervention for childhood obesity on dietary quality in a Chinese population, and to examine the mediating effect of dietary quality between intervention and changes in adiposity indicators.
I found the work well descripted and discussed, with a robust statistical analysis performed. I have only some comments to improve the manuscript
1) I suggest to improve introduction, in order to be more linear to improve the readability of the study. In addition, I suggest to underline the differences between the present study and the previously published in JAMA Pediatrics, in discussion or in introduction section.
2) In the abstract the intervention is missing. Please provide in brief
3) The abbreviation of SSB should be descripted also in the main text at the first mention, not only in the abstract
4) I suggest to add p value in all the tables (i.e. Table 1 and Table S3). Please provide
5) Typing error line 231
6) Line 238, p value is missing. Please provide.
Congratulations
Author Response
We would like to thank you for their valuable time and helpful comments.
Review1
1. I suggest to improve introduction, in order to be more linear to improve the readability of the study. In addition, I suggest to underline the differences between the present study and the previously published in JAMA Pediatrics, in discussion or in introduction section.
Response: Thank you for pointing this out. We have revised the introduction part. To point out the differences between current study and the publised primary study, we have added content in Page 2, line 71.
In the abstract the intervention is missing. Please provide in brief
Response: Thanks a lot for your advice. We have provided brief introduction of the intervention in the abstract. See Page 1, line 22-23.
- The abbreviation of SSB should be descripted also in the main text at the first mention, not only in the abstract
Response: Thank you for your kind remind. We have revised it.
- I suggest to add p value in all the tables (i.e. Table 1 and Table S3). Please provide
Response: Thank you for your suggestions. We have added P values in Table 1 and Table S3.
5.Typing error line 231
Response: Thank you for your kind remind. We have corrected the error.
6.Line 238, p value is missing. Please provide.
Response: Thank you for this comment. The P value has been added.
Reviewer 2 Report
This is a secondary analysis of the effect of DECIDE-Children on dietary quality, and its mediating effect on previously documented changes in BMI. This manuscript addresses interesting and important issues in a sophisticated way. The manuscript lacks some detail. The authors need to be more modest in what they claim DECIDE-Children attained.
The cluster randomized control design should be clearly stated in the abstract.
The reasonably large numbers of schools and students is admirable.
The very low percentages of overweight and obesity from 1992 to 2019 in China, compared to the West, suggest the nature of obesity may be different than in the West, and the methods of effective intervention may be different. In the Introduction it would be helpful to differentiate between documented influences on diet and obesity in China vs. the West, and point out documented commonalities.
The analysis model needs to include random terms for school and region (3) and a fixed effect for district (urban-rural). It is not clear to me this was the case.
Did the parents provide informed consent?
How were children recruited? What percent of children in the schools did not participate? Were there demographic differences between the children who participated and those who did not? Were the selected schools and their students representative of the children in their districts, as determined by district census data?
The manuscript would benefit from a process evaluation of dietary intervention components. This could be accomplished by inserting in a table each of the intervention procedures (for students, families, and schools) (e.g. number of books to be distributed, number of classes to be delivered, number of MVPA sessions delivered for one hour, etc.) with an indication of the extent to which compliance with each procedure was attained (e.g. percents). (School could be the unit of analysis.) This would allow an assessment of dose delivered by intervention level, which could also be related to outcomes.
How many items were in the FFQ? 12? That makes for large categories requiring students to make on-the-spot categorizations of the foods they usually consume, which may not correspond to the categorization of the authors. Please provide validity coefficients for the FFQ and DBI-Revision in comparable samples.
What were the categories for reporting the amount consumed per food group? What did the investigators do if a child consumed multiple items per category, which may vary in suggested/common portion size?
How was BF% measured?
Did the analyses follow intention to treat principles? Should they?
Table 2: Why was logistic regression used? Wouldn’t there be more power to treat the variables as continuous scores?
A substantial number of statistical tests were employed. Should the authors correct for the number of tests, or employ a more conservative level of significance?
The level of changes documented in these analyses was small, and not adequate to solve the growing obesity epidemic by itself. The authors need to be more modest in touting the effects of DECIDE-Children.
Table 2 revealed few changes in dietary intake. Did the intervention target all components of dietary quality or just SSB and unhealthy snacks? If the intervention targeted all components, were the authors concerned at the few changes obtained?
What procedures in the intervention were adapted to the cultural and dietary needs and practices of Chinese 4th grade children? How might these have been different in Western interventions?
This manuscript reports an efficacy study, i.e. an intervention delivered under optimal circumstances. When scaling up the intervention to be delivered under more usual circumstances the effect sizes will decrease. Can the authors anticipate what problems will be faced in scaling up the intervention, and what might be done to minimize the effect reduction, or magnify the effect?
The information in Table 5.1 needs to be expanded to specify S. Michie’s Behavior Change Techniques employed in each section.
Author Response
We would like to thank you for your valuable time and helpful comments.
Review2
- The cluster randomized control design should be clearly stated in the abstract.
Response: Thank you for the suggestion. We have pointed out the design of cluster randomized control in the abstract. See Page 1, line 22-23.
- The reasonably large numbers of schools and students is admirable.
Response: Thank you for your admiration.
- The very low percentages of overweight and obesity from 1992 to 2019 in China, compared to the West, suggest the nature of obesity may be different than in the West, and the methods of effective intervention may be different. In the Introduction it would be helpful to differentiate between documented influences on diet and obesity in China vs. the West, and point out documented commonalities.
Response: Thank you for your suggestion and we agree with your opinions. Our research group has conducted a systematic review and found that multi-component interventions appeared to be more effective than single-component intervention and the effects of physical activity interventions including curricular sessions were stronger[1]. However, scarce number of studies including dietary improvement components limited subgroup analyses and comparisons between China and the western countries. Previous evidence also had inconsistent conclusions on diet’s contributions[2-5].
Regarding to commonalities, studies in China and abroad have provided an important foundation as significant associations between dietary quality indices and obesity were found[6-9]. Concurrent changes of dietary quality and adiposity indicators were observed in interventions from other countries[10-12]. However, direct evidence is scarce worldwide for the effect of dietary quality as an underlying mechanism (that is, mediator) of a successful intervention with favorable changes of adiposity indicators in children. Considering Chinese different dietary characteristics and patterns, it is necessary to study the mediating effect of dietary quality and add evidence from China.
We have added these contents in the introduction part.
- The analysis model needs to include random terms for school and region (3) and a fixed effect for district (urban-rural). It is not clear to me this was the case.
Response: Thanks for your comment. Referred to previous research methods[13,14], we included school-level random intercepts because it was the level at which the intervention was assigned and clustering of children within schools might lead to correlation.
Because region is not the level of intervention assignment, there might be not so much correlation of dietary changes within a region. But we do agree with you that eating behavior might be similar in the same region. Thus, we conducted subgroup analysis to compare the effects of intervention among different regions rather than just adjusted it as a random intercept in the model. This can more directly exclude the influence of regional similarity in diet.
For district, we also thought that it might be associated with childhood obesity. Thus, we added sensitivity analyses and included the fixed effect of district (see Table S4 and Table S5). After adjusting for district, the results remained consistent.
5.Did the parents provide informed consent?
Response: The enrolled schoolchildren and their parents all have signed the informed consent.
- How were children recruited? What percent of children in the schools did not participate? Were there demographic differences between the children who participated and those who did not? Were the selected schools and their students representative of the children in their districts, as determined by district census data?
Response: Thank you for pointing this out. We have published a detailed protocol to introduce our research plan[15]. A total of 24 primary schools were selected (8 schools in each region) and 1728 students were eligible. Then project staff issued informed consent to all students and their parents in the selected classes. 287 parents declined to participate in the program. Parents who have signed the informed consent and agreed to participate (n=1441) were required to complete a questionnaire about the health status of their children. If a parent reported one of the following conditions, his or her children were excluded: (1) medical history of heart disease, hypertension, diabetes, tuberculosis, asthma, hepatitis or nephritis; (2) obesity caused by endocrine diseases or side effects of drugs; (3) abnormal physical development like dwarfism or gigantism; (4) physical deformity such as severe scoliosis, pectus carinatum, limp, obvious O-leg or X-leg; (5) inability to participate in school sport activities; and (6) a loss in weight by vomiting or taking drugs during the past 3months. Finally, 33 students were further excluded because of their ineligible health status.
Before children and their parents signed the informed consent, we didn’t have the right to get their basic information. Thus, we could not compare the basic characteristics of the participants who agreed to participate (n=1441) with those who declined (n=287). Besides, we thought that the proportion of children who declined to participate (287 out of 1728 students) was relatively low, which unlikely affect the representativeness of the data and a lot.
We could not compare the students included with the district census data. This might be a limitation, but we excluded boarding schools and specialty schools for children with talents or minority ethnic groups and selected schools from urban and rural districts to improve representativeness.
- The manuscript would benefit from a process evaluation of dietary intervention components. This could be accomplished by inserting in a table each of the intervention procedures (for students, families, and schools) (e.g. number of books to be distributed, number of classes to be delivered, number of MVPA sessions delivered for one hour, etc.) with an indication of the extent to which compliance with each procedure was attained (e.g. percents). (School could be the unit of analysis.) This would allow an assessment of dose delivered by intervention level, which could also be related to outcomes.
Response: Thank you for your suggestion. We conducted process evaluation in the 12 intervention schools and found the intervention was implemented at 12 intervention schools with high fidelity to most of the components (See Table S1). However, we didn’t collect the data related to fidelity and compliance situation in the 12 control groups. Thus, we could not further analyze their influence of compliance indicators on the outcomes.
- How many items were in the FFQ? 12? That makes for large categories requiring students to make on-the-spot categorizations of the foods they usually consume, which may not correspond to the categorization of the authors. Please provide validity coefficients for the FFQ and DBI-Revision in comparable samples.
Response: Thank you for pointing this out. Our FFQ contained 12 food subgroups, but when it comes to specific questionnaire items, it actually contained 16 items: Flour products, rice, corns, tubers, light color vegetables, dark color vegetables, fruits, red meat, poultry and game, fish and shrimp, eggs, milk and dairy products, soybean and soybean products, drinking water, SSB and unhealthy snacks. However, when calculating the DBI scores, some items were required to be synthesized into one subgroup: for example, flour products, rice, corns and tubers were synthesized into a subgroup of cereal, light color vegetables and dark color vegetables were synthesized into a subgroup of vegetables.
This FFQ for children and adolescents has been used in published study in China[16]. Regarding to the FFQ’s validity, after adjusted for total energy and intra-individual variation, all nutrient intakes showed positive correlation between FFQ and 24-hour recalls (P<0.05). The adjusted correlations ranged from 0.27 (for vitamin A) to 0.53 (for zinc), with a mean of 0.38.
Although we couldn’t find a gold standard technique and conduct validity research on DBI-revision, the revision version was referred to published study in China based on the Chinese Dietary Guidelines and has been used in some studies in China.
- What were the categories for reporting the amount consumed per food group? What did the investigators do if a child consumed multiple items per category, which may vary in suggested/common portion size?
Response: Children could choose to fill in how many times they consumed every week or every day and the amount (gram for solid food and milliliter for liquid) they consumed each time. We provided the unified standard tableware diagram and illustrative food pictures to help a child estimate the intake of different items in the same category. Take “flour product” as an example, pictures marked “a bowl of noodles is equal to 80g flour”, “a large steamed bun is equal to 100g of flour”, “a pancake bun is equal to 100g of flour”. If a child ate a mixed dish difficult to measure, the project investigator would record the situation in details and help to make a uniform estimation.
- How was BF% measured?
Response: Body weight was measured with body component instrument (Tanita MC-780 MA). Before measurement, students were required to take off shoes and socks, hats coats and pendants on their necks, hands and feet. They were also required to remove and put away metal objects such as keys and mobile phones. Then they stood barefoot on the four electrodes with their arms naturally hanging down and their thumbs extended on the top of the electrodes in their hands.
- Did the analyses follow intention to treat principles? Should they?
Response: Thanks for your questions. In the published protocol[15], we pointed out that sensitivity analysis would be performed on the primary outcome using imputation if the percentage of missing data exceeds 5%. In the primary study, thirty of 1392 children (2.2%) were lost to follow-up by the end of the trial. Thus, in current study, participants without data at the end of the trial were excluded in the main analysis according to the prespecified protocol. To improve the persuasiveness of our results, we also conducted sensitivity analyses following intention to treat principle by imputing the missing values (lost to follow-up) of dietary outcomes at the end of the trial using corresponding baseline values. The results remained consistent after imputation (see Table S4 and Table S5).
- Table 2: Why was logistic regression used? Wouldn’t there be more power to treat the variables as continuous scores?
Response: Thank you for this specific comment. We also initially analyzed subgroup scores as a continuous variable. However, the change of score were distributed discretely (i.e., figure below was the density distribution of change of scores of fruits) and the R2 of the model fitted was very low. Considering the premise of linear regression (the dependent variable should be normal distributed or approximate normal distributed), we changed the score into binary variable to improve reliability of the results.
- A substantial number of statistical tests were employed. Should the authors correct for the number of tests, or employ a more conservative level of significance?
Response: Thanks for your suggestion. The Benjamini-Hochberg procedure, which was thought to yield much greater power than the widely used Bonferroni technique, was added to control the false-discovery rate for multiple comparisons for each outcome[17]. After correcting, the effects on SSB intake and HBS scores were still significant. However, the effect on unhealthy snacks turned to be insignificant after correction. We have revised the related content in the whole manuscript.
- The level of changes documented in these analyses was small, and not adequate to solve the growing obesity epidemic by itself. The authors need to be more modest in touting the effects of DECIDE-Children.
Response: Thanks for your great suggestion. We agree that the level of changes was not that big in this research and the conclusion should be more modest. We have revised the whole manuscript. However, as written in the discussion, we also thought that although the effect size was small, childhood is an important period of habit formation and behavior change in this period might be paramount and accumulate over time to exert larger effects.
- Table 2 revealed few changes in dietary intake. Did the intervention target all components of dietary quality or just SSB and unhealthy snacks? If the intervention targeted all components, were the authors concerned at the few changes obtained?
Response: The intervention included instructions on all components because the recommended amount of these foods was clearly written in the Food Guidelines in China. But to be more specific, we set “not drinking sugar-sweetened beverage and eating less high-energy food” as our core information emphasized in the health education class.
We also agree that few changes in other food groups obtained needed further exploration. Previous studies also found that general messages are not likely to change specific non-targeted behaviors[18] and intake of vegetables and fruits was hard to improve in children[19]. We added in the discussion part that future research could try to enhance children’s dietary quality through methods like peer education, giving more specific instructions and improving canteen food supply. (See paragraph 2 in discussion)
- What procedures in the intervention were adapted to the cultural and dietary needs and practices of Chinese 4th grade children? How might these have been different in Western interventions?
Response: Thanks for your question. We thought the intervention had the following characteristics:
- As the previous study pointed out, prevention interventions appeared to be more effective in country like China where the obesity epidemic was at an earlier stage and fewer existing initiatives have been implemented by government, local authorities or schools[14].
- In China, most primary schools are government-funded public schools, policies such as not selling sugary drinks and snacks in schools do not affect school revenues a lot and were easy to generalize. But in western countries, after eliminating vending machines and stores, many private schools would struggle with maintaining profit margins and finding suitable fundraising alternatives[20].
- The intervention was also novel in its improvement of parental engagement. In China, parents' meetings are held regularly in schools and Wechat group were set for parents and schoolteachers, which help to give chances to deliver health education. It may be common in China for parents to give more attention to advice from schoolteachers than from other social institutions, which may be different to other countries[21].
- The Chinese Dietary Guidelines specify the intake range of all foods recommended, but in Western interventions, Dietary Guidelines (such as the US[22] and UK[23]) provide recommended serves rather than specific portion sizes.
We also added these contents in the discussion part. (See paragraph 2 in discussion)
- This manuscript reports an efficacy study, i.e. an intervention delivered under optimal circumstances. When scaling up the intervention to be delivered under more usual circumstances the effect sizes will decrease. Can the authors anticipate what problems will be faced in scaling up the intervention, and what might be done to minimize the effect reduction, or magnify the effect?
Response: Thank you for your question. We think that when scaling up the intervention the following problems could be done to minimize the effect reduction.
- One challenge was obesity prevention has not attracted enough attention from schools as greater attention was paid on academic performance by schools and families. The advancement of the State Council’s Action Plan for Healthy China, which included nutrition and health education for children, will help to promote widespread health education in school[24].
- Another challenge remaining was to how to fully engage family support in school-based interventions. Parental involvement in childhood obesity interventions is important as children are highly influenced by family unit, but some school-based obesity interventions showed difficulties in facilitating parental engagement. In the future, schools could consider various approaches, such as holding interactive classes and parent meetings, assigning homework, and monitoring BMI via the mobile health technology, to improve the attention of parents to obesity prevention and blend cooperation of family and school.
As you suggested, we further discussed these questions in the discussion section. (See paragraph 4 in discussion)
18.The information in Table 5.1 needs to be expanded to specify S. Michie’s Behavior Change Techniques employed in each section.
Response: Thanks for your suggestion. If we understand correctly, you are referring to Table S1. We have specified S. Michie’s Behavior Change Techniques employed in each section. (See Table S1)
Reference
- Liu, Z.; Xu, H.-M.; Wen, L.-M.; Peng, Y.-Z.; Lin, L.-Z.; Zhou, S.; Li, W.-H.; Wang, H.-J. A systematic review and meta-analysis of the overall effects of school-based obesity prevention interventions and effect differences by intervention components. The international journal of behavioral nutrition and physical activity 2019, 16, 95-95, doi:10.1186/s12966-019-0848-8.
- Brown, T.; Moore, T.H.; Hooper, L.; Gao, Y.; Zayegh, A.; Ijaz, S.; Elwenspoek, M.; Foxen, S.C.; Magee, L.; O'Malley, C.; et al. Interventions for preventing obesity in children. Cochrane Database Syst Rev 2019, 7, Cd001871, doi:10.1002/14651858.CD001871.pub4.
- Norman, G.J.; Kolodziejczyk, J.K.; Adams, M.A.; Patrick, K.; Marshall, S.J. Fruit and vegetable intake and eating behaviors mediate the effect of a randomized text-message based weight loss program. Prev Med 2013, 56, 3-7, doi:10.1016/j.ypmed.2012.10.012.
- Hammersley, M.L.; Okely, A.D.; Batterham, M.J.; Jones, R.A. Investigating the mediators and moderators of child body mass index change in the Time2bHealthy childhood obesity prevention program for parents of preschool-aged children. Public Health 2019, 173, 50-57, doi:10.1016/j.puhe.2019.04.017.
- Yιldιrιm, M.; Singh, A.S.; te Velde, S.J.; van Stralen, M.M.; MacKinnon, D.P.; Brug, J.; van Mechelen, W.; Chinapaw, M.J. Mediators of longitudinal changes in measures of adiposity in teenagers using parallel process latent growth modeling. Obesity (Silver Spring) 2013, 21, 2387-2395, doi:10.1002/oby.20463.
- Lioret, S.; McNaughton, S.A.; Cameron, A.J.; Crawford, D.; Campbell, K.J.; Cleland, V.J.; Ball, K. Three-year change in diet quality and associated changes in BMI among schoolchildren living in socio-economically disadvantaged neighbourhoods. The British journal of nutrition 2014, 112, 260-268, doi:10.1017/s0007114514000749.
- Berz, J.P.B.; Singer, M.R.; Guo, X.; Daniels, S.R.; Moore, L.L. Use of a DASH Food Group Score to Predict Excess Weight Gain in Adolescent Girls in the National Growth and Health Study. Archives of Pediatrics & Adolescent Medicine 2011, 165, 540-546, doi:10.1001/archpediatrics.2011.71.
- Okubo, H.; Crozier, S.R.; Harvey, N.C.; Godfrey, K.M.; Inskip, H.M.; Cooper, C.; Robinson, S.M. Diet quality across early childhood and adiposity at 6 years: the Southampton Women's Survey. International journal of obesity (2005) 2015, 39, 1456-1462, doi:10.1038/ijo.2015.97.
- Duan, R.; Liu, Y.; Xue, H.; Yang, M.; Cheng, G. [Cross-sectional association between overall diet quality and overweight/obesity among children and adolescents in Chengdu]. Zhonghua liu xing bing xue za zhi = Zhonghua liuxingbingxue zazhi 2014, 35, 994-998.
- Lee, S.Y.; Kim, J.; Oh, S.; Kim, Y.; Woo, S.; Jang, H.B.; Lee, H.J.; Park, S.I.; Park, K.H.; Lim, H. A 24-week intervention based on nutrition care process improves diet quality, body mass index, and motivation in children and adolescents with obesity. Nutrition research (New York, N.Y.) 2020, 84, 53-62, doi:10.1016/j.nutres.2020.09.005.
- Wadolowska, L.; Hamulka, J.; Kowalkowska, J.; Ulewicz, N.; Hoffmann, M.; Gornicka, M.; Bronkowska, M.; Leszczynska, T.; Glibowski, P.; Korzeniowska-Ginter, R. Changes in Sedentary and Active Lifestyle, Diet Quality and Body Composition Nine Months after an Education Program in Polish Students Aged 11⁻12 Years: Report from the ABC of Healthy Eating Study. Nutrients 2019, 11, 331, doi:10.3390/nu11020331.
- De Miguel-Etayo, P.; Moreno, L.A.; Santabárbara, J.; Martín-Matillas, M.; Azcona-San Julian, M.C.; Marti Del Moral, A.; Campoy, C.; Marcos, A.; Garagorri, J.M. Diet quality index as a predictor of treatment efficacy in overweight and obese adolescents: The EVASYON study. Clinical nutrition (Edinburgh, Scotland) 2019, 38, 782-790, doi:10.1016/j.clnu.2018.02.032.
- Liu, Z.; Gao, P.; Gao, A.Y.; Lin, Y.; Feng, X.X.; Zhang, F.; Xu, L.Q.; Niu, W.Y.; Fang, H.; Zhou, S.; et al. Effectiveness of a Multifaceted Intervention for Prevention of Obesity in Primary School Children in China: A Cluster Randomized Clinical Trial. JAMA pediatrics 2021, e214375, doi:10.1001/jamapediatrics.2021.4375.
- Li, B.; Pallan, M.; Liu, W.J.; Hemming, K.; Frew, E.; Lin, R.; Liu, W.; Martin, J.; Zanganeh, M.; Hurley, K.; et al. The CHIRPY DRAGON intervention in preventing obesity in Chinese primary-school--aged children: A cluster-randomised controlled trial. PLoS Med 2019, 16, e1002971, doi:10.1371/journal.pmed.1002971.
- Liu, Z.; Wu, Y.; Niu, W.-Y.; Feng, X.; Lin, Y.; Gao, A.; Zhang, F.; Fang, H.; Gao, P.; Li, H.-J.; et al. A school-based, multi-faceted health promotion programme to prevent obesity among children: protocol of a cluster-randomised controlled trial (the DECIDE-Children study). BMJ Open 2019, 9, e027902-e027902, doi:10.1136/bmjopen-2018-027902.
- Yan, Y.; Liu, J.; Zhao, X.; Cheng, H.; Huang, G.; Hou, D.; Mi, J.; China, C.; Adolescent Cardiovascular Health collaboration, g. Cardiovascular health in urban Chinese children and adolescents. Ann Med 2019, 51, 88-96, doi:10.1080/07853890.2019.1580383.
- Thissen, D.; Steinberg, L.; Kuang, D. Quick and Easy Implementation of the Benjamini-Hochberg Procedure for Controlling the False Positive Rate in Multiple Comparisons. Journal of Educational and Behavioral Statistics 2002, 27, 77-83, doi:10.3102/10769986027001077.
- Dixon, L.B.; Tershakovec, A.M.; McKenzie, J.; Shannon, B. Diet quality of young children who received nutrition education promoting lower dietary fat. Public health nutrition 2000, 3, 411-416, doi:10.1017/s1368980000000471.
- Evans, C.E.; Christian, M.S.; Cleghorn, C.L.; Greenwood, D.C.; Cade, J.E. Systematic review and meta-analysis of school-based interventions to improve daily fruit and vegetable intake in children aged 5 to 12 y. The American journal of clinical nutrition 2012, 96, 889-901, doi:10.3945/ajcn.111.030270.
- Mâsse, L.C.; Naiman, D.; Naylor, P.J. From policy to practice: implementation of physical activity and food policies in schools. The international journal of behavioral nutrition and physical activity 2013, 10, 71, doi:10.1186/1479-5868-10-71.
- Li, Q.; Guo, L.; Zhang, J.; Zhao, F.; Hu, Y.; Guo, Y.; Du, X.; Zhang, S.; Yang, X.; Lu, C. Effect of School-Based Family Health Education via Social Media on Children’s Myopia and Parents’ Awareness: A Randomized Clinical Trial. JAMA Ophthalmology 2021, 139, 1165-1172, doi:10.1001/jamaophthalmol.2021.3695.
- Cohen, J.F.W.; Kraak, V.I.; Choumenkovitch, S.F.; Hyatt, R.R.; Economos, C.D. The CHANGE study: a healthy-lifestyles intervention to improve rural children's diet quality. J Acad Nutr Diet 2014, 114, 48-53, doi:10.1016/j.jand.2013.08.014.
- Robertson, W.; Fleming, J.; Kamal, A.; Hamborg, T.; Khan, K.A.; Griffiths, F.; Stewart-Brown, S.; Stallard, N.; Petrou, S.; Simkiss, D.; et al. Randomised controlled trial evaluating the effectiveness and cost-effectiveness of 'Families for Health', a family-based childhood obesity treatment intervention delivered in a community setting for ages 6 to 11 years. Health Technol Assess 2017, 21, 1-180, doi:10.3310/hta21010.
- Xu, H.; Ecker, O.; Zhang, Q.; Du, S.; Liu, A.; Li, Y.; Hu, X.; Li, T.; Guo, H.; Li, Y.; et al. The effect of comprehensive intervention for childhood obesity on dietary diversity among younger children: Evidence from a school-based randomized controlled trial in China. PloS one 2020, 15, e0235951, doi:10.1371/journal.pone.0235951.
Round 2
Reviewer 2 Report
The authors were responsive to this reviewer's comments.